Filtering out parasites: sand crabs (Lepidopa benedicti) are infected by more parasites than sympatric mole crabs (Emerita benedicti)

Faulkes Zen zen.faulkes@utrgv.edu
Department of Biology, The University of Texas Rio Grande Valley , Edinburg , TX , United States of America
Ford Alex
Electronic publication date: 2017 Sep 22
Publication date: 2017
Volume: 5
Electronic Location ID: e3852
Received 2017 Jul 15; Accepted 2017 Sep 5
Copyright: ©2017 Faulkes
Copyright year: 2017
Copyright holder: Faulkes
License: This is an open access article distributed under the terms of the Creative Commons Attribution License, which permits unrestricted use, distribution, reproduction and adaptation in any medium and for any purpose provided that it is properly attributed. For attribution, the original author(s), title, publication source (PeerJ) and either DOI or URL of the article must be cited.
License URL: https://creativecommons.org/licenses/by/4.0/

Keywords: Crustacean, Parasite manipulation of behavior, Nematode, Parasite, Cestode, Hippoidea, Digging

Funding: The author received no funding for this work.

==============================
Two digging decapod crustaceans, the sand crab species Lepidopa benedicti and the mole crab species Emerita benedicti, both live in the swash zone of fine sand beaches. They were examined for two parasites that infect decapod crustaceans in the region, an unidentified nematode previously shown to infect L. benedicti, and cestode tapeworm larvae, Polypocephalus sp., previously shown to infect shrimp (Litopenaeus setiferus). Lepidopa benedicti were almost always infected with both parasite species, while E. benedicti were rarely infected with either parasite species. This difference in infection pattern suggests that tapeworms are ingested during sediment feeding in L. benedicti, which E. benedicti avoid by filter feeding. Larger L. benedicti had more Polypocephalus sp. larvae. The thoracic ganglia, which make up the largest proportion of neural tissue, contained the largest numbers of Polypocephalus sp. larvae. Intensity of Polypocephalus sp. infection was not correlated with how long L. benedicti remained above sand in behavioural tests, suggesting that Polypocephalus sp. do not manipulate the sand crabs in a way that facilitates trophic transmission of the parasite. Litopenaeus setiferus may be a primary host for Polypocephalus sp., and L. benedict may be a secondary, auxiliary host.

Introduction

Parasites can be generalists that infect many host species, or specialists that infect only a small number of host species, or even just one host species. (Poulin, 2007; Schmid-Hempel, 2011; Loker & Hofkin, 2015). A benefit of being a specialist may be increased adaptation to a host species. Specialization should be favoured in endoparasites that manipulate host behaviour (Adamson & Caira, 2011; Fredensborg, 2014), because the nervous systems generating behaviour are probably anatomically and physiologically more variable than other types of tissue (Bullock, 1993; Bullock, 2004; Bullock, 2006). Behavioural manipulation often manifests as parasite induced tropic transmission (PITT), in which parasites with multiple host life cycles change the behaviour of one host in such a way as to enhance the likelihood of the host being eaten by a predatory species that is the next host in the parasite’s life cycle (Moore, 2002; Lafferty & Shaw, 2013).

Sand crabs (Lepidopa benedicti) and mole crabs (Emerita benedicti) are digging anomuran crustaceans in the same superfamily (Hippoidea) (Fig. 1), which are both found in the swash zones of sandy beaches in the Gulf of Mexico. Given that they are relatively closely related and found in the same habitat, it is a reasonable hypothesis that they might have similar parasites to each other. Lepidopa benedicti is often infected by an unidentified nematode species that does not appear to manipulate host behaviour (Joseph & Faulkes, 2014), which might also infect E. benedicti.

Figure 1 Digging crab species.

(A) Lepidopa benedicti. (B) Emerita benedicti.

Another parasite that infects decapod crustaceans where these two species live are larval cestode tapeworms in the genus Polypocephalus, which infect white shrimp (Litopenaeus setiferus) (Carreon, Faulkes & Fredensborg, 2011; Carreon & Faulkes, 2014). Although the life cycle of species in this genus is not completely worked out, it seems likely that it is a two part life cycle (Fig. 2): crustaceans (Villella, Iversen & Sindermann, 1970; Owens, 1985; Shields, 1992; Hudson & Lester, 1994; Brockerhoff & Jones, 1995; Payne, 2010) and other invertebrates (Cake Jr, 1979) for the larval stage, and elasmobranch fishes (e.g., skates and rays) as the definitive hosts for adults (Butler, 1987; Call, 2007; Koch, 2009). There are reasons that could suggest Polypocephalus spp. could be either generalists or specialists. On the one hand, Polypocephalus spp. larvae infect multiple species from at least two phyla (Cake Jr, 1979; Owens, 1985; Brockerhoff & Jones, 1995), suggesting that species in this genus are generalists. On the other hand, Polypocephalus sp. inhabit the nervous system of crustaceans, and appear to manipulate behaviour in L. setiferus (Carreon, Faulkes & Fredensborg, 2011), which are factors that suggest species in this genus are specialists.

Figure 2 Hypothesized life cycle of Polypocephalus sp.

Larval stages of cestode tapeworms in the genus Polypocephalus infect crustaceans and other invertebrates. These intermediate hosts are presumably ingested by the putative definitive hosts, skates and rays, which are expected to excrete Polypocephalus eggs. Images from the Noun Project https://thenounproject.com: shrimp by Jeffrey Qua, crab by Mallory Hawes, scallop by B Barrett, and skate by Örn Smári Gíslason, used under CC BY 3.0 license https://creativecommons.org/licenses/by/3.0/us/.

Polypocephalus sp. is also a candidate for studying the manipulation of host behaviour, because the larval stage infects the neural tissue of their decapod crustacean hosts. Being in or near the nervous system would seem to make such manipulation easier for parasites. In white shrimp, increased infection was correlated with increased activity of the host (Carreon, Faulkes & Fredensborg, 2011), which was hypothesized to be a case of parasite-induced trophic transmission. A trophically transmitted parasite in a digging crustacean might be excepted to change the behaviour of its host so it spends more time above sand (Joseph & Faulkes, 2014). Litopenaeus setiferus do dig into sand (Eldred et al., 1961; Fuss Jr, 1964; Pinn & Ansell, 1993), and their increased activity with infection would be consistent with the prediction above.

This paper compares the patterns of infection in L. benedicti and E. benedicti for both nematode and cestode parasites, and tests whether Polypocephalus sp. manipulates the behaviour of L. benedicti as they do with shrimp (Carreon, Faulkes & Fredensborg, 2011).

Methods

Sand crabs (Lepidopa benedicti) and mole crabs (Emerita benedicti) were collected from the beaches of South Padre Island, Texas by turning over sand with a shovel near, and parallel to, the waterline of the shore (Faulkes, 2017; Murph & Faulkes, 2013). Crabs found in the overturned sand or in the water of the trench created were collected. Individuals were sexed by examining pleopod size (longer in females) and the carapace length was measured with digital calipers. Different individuals were used to study infection by nematodes and Polypocephalus sp. To examine infection of nematodes, E. benedicti were broken using forceps, and nematodes found in the dissecting dish were counted, following the previous study on L. benedicti (Joseph & Faulkes, 2014). To examine infection of Polypocephalus sp., individuals were anaesthetised by chilling for ∼20 min on crushed ice, dissected in sea water, and the nerve cord was removed. The nerve cord was cut into smaller sections, which were pinned in dishes lined with Sylgard (Dow Corning, Midland, MI, USA). The nerve cords were dehydrated in a progressive ethanol series (70% ethanol for 5 min, 90% ethanol for 5 min, 100% ethanol for 5 min, then 100% ethanol again for 10 min), cleared in methyl salicylate on a depression slide, viewed under a compound microscope (Olympus CX41), and photographed (Olympus C-5050 Zoom digital camera), following a previous study of L. setiferus (Carreon & Faulkes, 2014). In some cases, consecutive images at different focal points in the Z axis were compiled into a single image using Helicon Focus v. 6.7.1 Lite (Helicon Soft Ltd., Kharkiv, Ukraine).

Initial observations of 10 individuals of each species indicated that variation in numbers of parasites infecting L. benedicti was sufficient to test whether there was a correlation between infection and host behaviour. Because few E. benedicti were infected, and there was very little variation in the number of parasites of those that were infected, their behaviour was not examined.

Behavioural tests were similar to those described in Joseph & Faulkes (2014). Individuals were video-recorded digging in a tank 300 mm wide × 150 mm deep × 200 mm high, filled with ∼40 mm of sand from South Padre Island covered by ∼120 mm of seawater. Video was recorded with an iPad 3 using Coach My Video v. 4.4 (http://www.coachmyvideo.mobi). Individuals were released at the top of the tank, and were filmed until the carapace was covered by sand. The total time was calculated by subtracting the submergence times from release time (rounded down to whole seconds). Individuals made three digging trials, each separated by a 5 min rest period when the animal was not disturbed to minimize habituation. The average of the three trials was used for analysis.

The behaviour of crabs fell into three basic categories. An individual could (1) immediately dig into sand (“direct”), (2) stay above sand by tailflipping and rowing its legs (Faulkes & Paul, 1997) before digging (“swim”), or; (3) remain on the top of the sand, immobile, before digging (“sit”). “Swim” and “sit” are not mutually exclusive. An individual could do both in one trial, in either order. For simplicity of analyses, individuals that both swam and “sat” in their three trials were omitted from analyses that examined individuals.

Descriptive statistical analyses and graphs were made in Origin 2017 (OriginLab Corporation). Nonparametric tests were used for most analyses because of nonhomogenous variation in data distribution. Nonparametric statistical analyses were performed in SPSS v. 23 (IBM, Armonk, NY, USA).

Results

The previously reported prevalence of nematodes in L. benedicti (87.0%, n = 46) (Joseph & Faulkes, 2014) was higher than in E. benedicti (0.0%, n = 22) (Fig. 3A). Similarly, the prevalence of Polypocephalus sp. infection in Lepidopa benedicti (98.0%, n = 50) was higher than in E. benedicti (18.2%, n = 22) (Fig. 3B). The mean intensity of Polypocephalus sp. infection (Figs. 3B and 4) was greater in L. benedicti (range = 1–170, mean = 34.5, SD = 33.0, n = 49; uninfected animals excluded) than E. benedicti (range = 1–3, mean = 1.5, SD = 1.0, n = 4; uninfected animals excluded). These differences are not because of the overall size of individuals examined: the average size of L. benedicti was smaller than E. benedicti (Table 1) in both cases. Because there were so few parasites of either species in E. benedicti, all further analyses concern only L. benedicti.

Polypocephalus sp. larvae were closely associated specifically with neural tissue, including peripheral nerves to appendages (Fig. 4). The larvae often appeared on the surface of ganglia and could sometimes be seen on the dissected nerve cord using a stereomicroscope.

There is a significant correlation (Spearman’s ρ = 0.49, p = 0.002, n = 38) between L. benedicti size and mean intensity of Polypocephalus sp. infection (Fig. 5).

Figure 3 Infection patterns of sand crabs and mole crabs by parasites.

(A) Infection of crabs by unidentified nematode species. Lepidopa benedicti data redrawn from Joseph & Faulkes (2014). (B) Infection pattern of crabs by Polypocephalus sp. larvae. Summary statistics: square, mean; line dividing box, median; box, 50% of data; whiskers, 95% of data; triangles, minimum and maximum. Raw data shown by dots.

Figure 4 Micrographs of Polypocephalus sp. in nervous tissue.

(A) Lepidopa benedicti thoracic ganglion 2. (B) Lepidopa benedicti thoracic ganglia 3, and fused ganglion consisting of thoracic ganglia 4, 5, and abdominal ganglion 1. Different individual than (A). (C) Emerita benedicti brain. Arrow indicates single Polypocephalus larvae. (D) Emerita benedicti thoracic ganglion 1. No Polypocephalus sp. larvae in this individual. Different individual than (C). Anterior towards top in (A), and towards left in (B–D).

Figure 5 Bigger Lepidopa benedicti have more Polypocephalus sp. larvae.

Relationship between size of L. benedicti and intensity of Polypocephalus sp. infection.

Table 1 Size of animals used in study.

	Lepidopa benedicti	Emerita benedicti	
Parasite	Mean carapace length	SD	n	Mean carapace length	SD	n	
Nematode sp.	11.44 mm	2.83	46	13.44 mm	5.45	22	
Polypocephalus sp.	9.95 mm	1.72	49	18.19 mm	3.65	21	

Like other anomurans, L. benedicti have shorter abdomens than familiar decapods like shrimp and crayfish. Because L. benedicti are specialized for digging and swimming with thoracic legs 1 through 4, the legs are proportionately more robust. Thoracic leg 5 is very small and used for grooming. These anatomical features are reflected in the relative sizes of the ganglia in L. benedicti compared to other decapod crustaceans. The thoracic ganglia associated with thoracic legs 1–4 are substantially larger than abdominal ganglia 2–6. The fourth and fifth thoracic ganglia and the first abdominal ganglion are fused. The number of larvae in the ganglia differed significantly across the nervous system (Kruskal Wallis = 16.71, df = 6, p = 0.01), with thoracic ganglia containing the most larvae, particularly in highly infected individuals (Fig. 6).

Figure 6 Number of Polypocephalus sp. larvae in different regions of the nerve cord in Lepidopa benedicti.

Summary statistics: square, mean; line dividing box, median; box, 50% of data; whiskers, 95% of data; triangles, minimum and maximum. Raw data shown by dots. Groups sharing a letter do not differ significantly from each other. Sample sizes vary because some ganglia were damaged during dissection. SEG, subesophageal ganglion; T, thoracic ganglion; A, abdominal ganglion.

Contrary to the prediction that more heavily infected animals would spend more time above sand, the mean intensity of Polypocephalus sp. infection was not significantly correlated (Spearman’s ρ =  − 0.233, p = 0.16, n = 38) with mean digging time (Fig. 7). Size of L. benedicti was not significantly correlated (Spearman’s ρ =  − 0.279, p = 0.09, n = 38) with mean digging time (Fig. 8), confirming previous findings (Joseph & Faulkes, 2014).

Figure 7 Polypocephalus sp. infection does not affect speed of digging in Lepidopa benedicti.

Relationship between intensity of Polypocephalus sp. infection and digging time in L. benedicti.

Figure 8 Size does not affect digging time in Lepidopa benedicti.

Relationship between carapace length and mean digging time in L. benedicti.

The three main behaviours of L. benedicti (directly digging into sand, swimming, or remaining stationary, or “sitting”) were significantly different (Kruskal-Wallis = 70.76, df = 2, p < 0.01) in how long individuals remained above sand (Fig. 9). Swimming above sand and remaining stationary on top of it did not differ significantly in the duration of exposure for sand crabs, although “sitting” times had greater variation, resulted in the longest times that sand crabs were exposed.

Figure 9 Time above sand for different behaviours by Lepidopa benedicti.

Duration of individual behaviour trials, grouped by different behaviours. N = 110 trials. Four trials involved combinations of sitting and swimming, and are not shown due to their rarity. Summary statistics: square, mean; line dividing box, median; box, 50% of data; whiskers, 95% of data; triangles, minimum and maximum. Raw data shown by dots. Groups sharing a letter do not differ significantly from each other.

Individuals showing different behaviour patterns had significantly different mean intensities of infection (Kruskal Wallis = 8.72, df = 2, p = 0.013): animals that “sat” at least once had lower infection intensities than those that swam at least once or always dug directly into sand (Fig. 10).

Figure 10 Infection intensity of Lepidopa benedicti individuals showing different behaviours.

Individuals categorized into three groups: those that always dug directly; those that swam at least once, but never “sat” (i.e., remaining immobile on the surface); those that “sat” at least once, but never swam. Summary statistics: square, mean; line dividing box, median; box, 50% of data; whiskers, 95% of data; triangles, minimum and maximum. Raw data shown by dots. Groups sharing a letter do not differ significantly from each other.

Discussion

Two parasite species, an unidentified nematode (Joseph & Faulkes, 2014) and Polypocephalus sp., infect Lepidopa benedicti with much higher prevalence and intensity than in Emerita benedicti. In the case of Polypocephalus sp., a high prevalence and intensity also occurs in white shrimp (L. setiferus) which also dig into sand (Eldred et al., 1961; Fuss Jr, 1964; Pinn & Ansell, 1993). What distinguishes E. benedicti from both L. benedicti and L. setiferus is the feeding mode. Emerita species are filter feeders (Efford, 1966), which L. benedicti and L. setiferus are not. Lepidopa species are probably sediment feeders (Boyko, 2002). This suggests that ingestion is a common route of Polypocephalus sp. infection for L. benedicti and L. setiferus. Presumably, E. benedicti avoid infection because they are filtering food from the water column, which is hypothesized to have extremely low numbers of Polypocephalus sp. cysts compared to sand and other surfaces.

The lack of parasites in E. benedicti in this population is unusual not only because the sympatric L. benedicti is infected, but because other populations of Emerita species are infected with other parasites (Smith, 2007; Oliva et al., 2008; Kolluru et al., 2011; Violante-Gonzalez et al., 2015; Violante-González et al., 2016). Because this study did not run an entire year, however, it is possible that E. benedicti infections vary seasonally or spatially, and that this species is heavily infected at other times or places. That this study did not address whether there any substantial variation in infection rates over time means that the differences in nematode infection in L. benedicti and E. benedicti should be interpreted with cautiously, because the data for L. benedicti (Joseph & Faulkes, 2014) was collected before the data for E. benedicti (this study).

In L. setiferus, the greatest number of Polypocephalus sp. larvae is in the abdominal ganglia (Carreon, Faulkes & Fredensborg, 2011), but in L. benedicti, the greatest number is in the thoracic ganglia. This probably reflects which region has the proportionately greater mass of neural tissue available in the two species, although neural mass does not entirely explain distribution patterns across the nervous system (Carreon & Faulkes, 2014). Another difference is that in L. setiferus, Polypocephalus sp. larvae appeared to be more deeply embedded in neural tissue and were rarely visible under a dissecting microscope until the nerve cord was either squashed or cleared. In L. benedicti, larvae were in comparatively superficial positions, and could be seen with dissecting microscopes. There also appeared to be less variation in Polypocephalus sp. larval size in L. setiferus than L. benedicti (compare Fig. 4 here to Fig. 1 in Carreon, Faulkes & Fredensborg, 2011).

Polypocephalus sp. does not seem to manipulate L. benedicti in a way that would facilitate trophic transmission. Intuitively, one would predict that if Polypocephalus sp. were manipulating sand crabs to make them vulnerable to predators, animals with more Polypocephalus sp. would be more likely to swim or remain immobile on the top of the sand. In anything, the evidence points towards more heavily infected individuals being more likely to dig into sand immediately. Nevertheless, this result can be viewed as consistent with the results in L. setiferus, where higher levels of infection increased activity (Carreon, Faulkes & Fredensborg, 2011). Digging directly into sand and swimming could both be considered higher activity by L. benedicti.

The apparent difference in parasite-induced behavioural manipulation in L. setiferus and L. benedicti has several potential explanations. First, the Polypocephalus species infecting L. setiferus may not be the same species as the one infecting L. benedicti. Although both dig in sand, there are differences in the life history of the two hosts. For example, L. setiferus transition from living in seagrass beds (Zimmerman & Minello, 1984) to deeper water as they grow, and change preferences for salinity over their lives (Williams, 1984), whereas L. benedicti settle into sand after metamorphosing from a pelagic larva and remain there for their entire lives (Stuck & Truesdale, 1986). These differences in the niches of the host species could be consistent with there being multiple Polypocephalus species. Genetic testing will eventually be able to determine if there is one cestode species or multiple species. Second, L. setiferus may be the preferred primary host for Polypocephalus sp. (perhaps along with other shrimp species), and L. benedicti is a non-preferred auxiliary host. The intensity of Polypocephalus sp. larvae in L. setiferus (mean = 97.7, SD = 102.6; maximum 397; n = 53; Carreon, Faulkes & Fredensborg, 2011) is approximately triple that of L. benedicti (mean = 34.5, SD = 33.0; maximum 170; n = 49; this study). Litopenaeus setiferus may be more abundant than L. benedicti. Litopenaeus setiferus is commercially fished, and annual catches from trawling in the Texas waters of Gulf of Mexico average 7 million pounds per year (Texas Parks and Wildlife, 2002). In contrast, 10 m transects of beach often yield less than 10 L. benedicti individuals (Faulkes, 2017; Murph & Faulkes, 2013). L. benedicti populations have only been sampled in the swash zone (Faulkes, 2017; Faulkes, 2014) and its abundance in deeper waters is unknown (it has been recorded up to 60 m depth; Boyko, 2002). Nevertheless, it seems plausible that the biomass for L. benedicti, and thus its potential as host for Polypocephalus sp., is lower than L. setiferus. Thus, there may be greater selection pressure for Polypocephalus sp. to manipulate its primary host but not secondary ones.

Meera Joseph contributed data for Emerita benedicti nematode infection.

Additional Information and Declarations

Competing Interests

Author Contributions

Data Availability

The author declares there are no competing interests.

Zen Faulkes conceived and designed the experiments, performed the experiments, analyzed the data, contributed reagents/materials/analysis tools, wrote the paper, prepared figures and/or tables, reviewed drafts of the paper.

The following information was supplied regarding data availability:

Faulkes, Zen (2017): Polypocephalus infection of hippoid crabs. figshare. https://doi.org/10.6084/m9.figshare.5183515.v1.

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
