# Peer review of "Filtering out parasites: sand crabs (Lepidopa benedicti) are infected by more parasites than sympatric mole crabs (Emerita benedicti)"

_PeerJ, doi:10.7717/peerj.3852_

## Round 0.1 · original submission · Major Revisions

Dear Dr Fawlkes,

The reviewers have no had time to comment on your manuscript and whilst they found merit in its publication they have also outlined a number of corrections they feel you should address. We look forward to receiving a revised manuscript from you and thank you for submitting your paper to PeerJ.

[# Staff Note: It is PeerJ policy that additional references suggested during the peer-review process should only be included if the authors are in agreement that they are relevant and useful #]

Reviewer 1 ·

Basic reporting

In the first line of the Introduction (L12) – the phrasing is rather awkward and should be changed to read “Parasites can be generalists that infect many different host species, or specialists that only infect a small number or even a single host species.” Additionally, the author should also provide a general reference for this statement, which can be one of the following:

Poulin, R. (2006). Evolutionary ecology of parasites. 2nd ed. Princeton University Press.

Schmid Hempel, P., & Schmid-Hempel, P. (2011). Evolutionary parasitology: the integrated study of infections, immunology, ecology, and genetics. Oxford University Press

Loker, E., & Hofkin, B. (2015). Parasitology: a conceptual approach. Garland Science.

At L25 poor reasoning was given for why that the unidentified nematode species might be a generalist. There are many non-host manipulating parasites which are specialists that infect only a few or even a single species. This should be modified to provide a better explanation for why the nematode might be a generalist.

Experimental design

No comment

Validity of the findings

In the Result section at L100, it would be very helpful for the read to also report the range (minimum – maximum) of parasite intensity. While this could be worked out by looking at Figure 2, this is not as exact as if actual numbers are provided.

In the Discussion section at L146, the author stated “E. benedicti in this population is usual not only because…” – did they actually meant “unusual”?

I have a suggestion about regarding the different patterns of cestode and nematode infection in the two crustacean species. Could there be some differences in the behaviour or other aspects of the two species’ ecology which makes one more susceptible or exposed to acquiring the infective stages of the parasites? This is especially worth considering since the unidentified parasitic nematode was absent in E. benedicti, which also infected far less frequently by Polycephalus larvae and in much lower numbers. For example, see:

Koehler, A. V., Gonchar, A. G., & Poulin, R. (2011). Genetic and environmental determinants of host use in the trematode Maritrema novaezealandensis (Microphallidae). Parasitology, 138(1), 100-106.

Differences in the body size of the two species may also be a contributing factor to why one species is more heavily (and in the case of the larval nematodes, just one of the species was infected) than the other. For example, see:

Leung, T. L., & Poulin, R. (2008). Size-dependent pattern of metacercariae accumulation in Macomona liliana: the threshold for infection in a dead-end host. Parasitology research, 104(1), 177-180.

Additional comments

The findings of this study are interesting and scientifically robust. The analyses used were appropriate and the interpretation of the findings was generally sound, therefore, once the author has made the suggested changes, it should be accepted for publications in PeerJ.

·

Basic reporting

This is a nice little study consisting of two parts. Firstly, parasite abundance was compared between two co-occurring sand crab species. Secondly, behavioral tests were conducted on one of them to investigate whether the cestode Polypocephalus sp. induces behavioral changes, which may indicate parasite induced trophic transmission to the final host. This is of particular interest because the parasite has been linked to behavioral changes in a different crustacean (Carreon et al. 2011), and the behavioral tests on Lepidopa benedicti and the quantification of parasites in that species provide the main story of this paper.
There are some issues related to the comparison of parasite data between the two species, and there are some potential issues related to the interpretations of the behavioral data, which I will address in more detail in the relevant sections of my review below.
1) The study system is well introduced. However, the concept of PITT deserves to be defined and introduced earlier in the Introduction followed by a general reference. In this case, Lafferty and Shaw, 2013 would be a suitable reference. Moore, 2002 is another good general reference (book).
2) Line 22-23: I think it is reasonable to assume that species sharing the same habitat are exposed to the same set of parasites. However, there could be many physiological and behavioral reasons why some parasites successfully infect one species and not another species. Therefore, it is a stretch to assume that co-occurring species harbor the same set of parasites. I recommend that you rephrase the sentence accordingly. I think it is important to not produce a too simplistic view of host specificity. Many other factors than the ability to change host behavior could be important. For example, parasites, which have to negotiate a potentially strong immune response are often host specific.
3) Be clear and consistent in the use of terminology. Parasite intensity refers to the number of parasites in infected hosts only. If uninfected individuals are included (which I believe you do), then use the term parasite abundance, which includes all individuals examined (including uninfected individuals). Refer to Bush et al. 1997 for a more detailed outline of different parasitology terminology.
4) L. 108-110: This sentence belongs in the Discussion (not part of this study).
Figures
5) In Figure 3 I believe that the scale bar is incorrect in either A, or in B and C (the size of the parasite is very different between photos). Also in Fig. 3, I would only provide two photos, one from Lepidopa benedicti and one from Emerita benedicti, each of them indicating parasites in tissue by arrows.
6) Fig. 5. In the Figure legend I suggest that you define the abbreviated term SEG. Sub-esophageal ganglion may not be known to a wider audience.
7) There are quite a few typos/spelling mistakes throughout the manuscript, which need to be addressed (e.g. lines 41, 57, 80, 114, 141, 146, 150, 160, 168, 169).

Experimental design

1) The research question states that behaviour would be tested in both species (L. 50-53). However, this was only done for Lepidopa benedicti. Rephrase the statement accordingly.
2) It is problematic that you compare nematode data collected at different times between the two sand crab species. This is simply not meaningful in many cases since parasite abundance may vary greatly both spatially (within even a few meters) and temporally. My recommendation is that you remove data on the unidentified nematode (Fig. 2). You can mention in the Results section that you did not find the nematode in Emerita in this study, and then refer to the previously published study reporting nematode infections in Lepidopa (Joseph and Faulkes 2014) as long as you do not directly compare the two findings.
3) The methods are not explained in sufficient detail for replication. For example, how and where specifically were the animals collected? (it only says that they were collected). Also, how were Emerita dissected to examine for the presence of nematodes?
4) L. 70-72: A minor comment, but behavior alterations may not necessarily be intensity-dependent. Therefore, presence/absence of a parasite may still provide an indication of if a parasite may change host behaviour.

Validity of the findings

1) My main concern regarding the interpretation of your results is that the effect of crab size on behavior was not investigated. You report that crab size was significantly and positively related to the number of Polypocephalus per individual (Fig. 4). That could also mean that the negative relationship between the number of Polypocephalus and digging time observed (Fig. 6+7) was influenced by the effect of crab size. Maybe you can use crab size as a co-variate in your analyses to correct for the potential confounding effect of size.
2) You use parametric tests to test the relationship btw size and parasite numbers, and digging time and parasite numbers. To me it looks like your data violate the assumptions of homogeneity of variance in parasite numbers (y values) across x values (size). It is clear that variance in parasite numbers increase with increasing size of the sand crabs. I therefore suggest using a Spearman rank test instead of a Pearson correlation or Linear regression analysis. You may also be able to transform your data so that you can still use a parametric test (especially if you want to include size as a covariate).
3) I wonder if you measured the dimensions of the larval Polypocephalus in any of the two host species or in white shrimp? Molecular techniques are necessary to determine the identity of Polypocephalus collected from different hosts, but morphological measurements may indicate obvious differences.
4) Along the lines of the point above, I think you should mention that white shrimp generally occupy a different habitat from sand crabs (sea grass beds as juveniles and deeper waters as adults in contrast to the swash zone of sandy beaches), which may suggest that we are dealing with a different species of Polypocephalus.

---

## Round 0.2 · accepted · Accept

Thank you for submitting a revised version of the manuscript. I am delighted to tell you your manuscript has been accepted for publication and wish to thank you for submitting your manuscript to PeerJ

Reviewer 1 ·

Basic reporting

I am satisfied with the revised version of the manuscript and recommend that it be accepted for publication.

Experimental design

no comment

Validity of the findings

no comment

·

Basic reporting

I have no further comments

Experimental design

No further comments

Validity of the findings

No further comments